# Graph Counterfactual Explainable AI via Latent Space Traversal

Andreas Abildtrup Hansen[*1], Paraskevas Pegios[1], Anna Calissano[2], and Aasa Feragen[1,3]

[1]Technical University of Denmark, Kongens Lyngby, Denmark
[2]Imperial College London, London, England
[3]Pioneer Centre for AI, Copenhagen, Denmark

## Abstract

Explaining the predictions of a deep neural network is a nontrivial task, yet high-quality explanations for predictions are often a prerequisite for practitioners to trust these models. *Counterfactual explanations* aim to explain predictions by finding the "nearest" in-distribution alternative input whose prediction changes in a pre-specified way. However, it remains an open question how to define this nearest alternative input, whose solution depends on both the domain (e.g. images, graphs, tabular data, etc.) and the specific application considered. For graphs, this problem is complicated i) by their discrete nature, as opposed to the continuous nature of state-of-the-art graph classifiers; and ii) by the node permutation group acting on the graphs. We propose a method to generate counterfactual explanations for any differentiable black-box graph classifier, utilizing a case-specific permutation equivariant graph variational autoencoder. We generate counterfactual explanations in a continuous fashion by traversing the latent space of the autoencoder across the classification boundary of the classifier, allowing for seamless integration of discrete graph structure and continuous graph attributes. We empirically validate the approach on three graph datasets, showing that our model is consistently high-performing and more robust than the baselines.

## 1 Introduction

Sets of graphs arise in different applications, e.g. brain connectivity [1–3], brain arterial networks [4], anatomical trees [5, 6], mobility networks [7], and chemistry, and graph classifiers play important roles in our daily infrastructure, healthcare quality, and national security. Their explainability is crucial to ensure both safety and trust when using AI for high-stakes applications.

However, explainable AI (XAI) for graph predictors remains challenging due to the misalignment between the discrete graph structure and the continuous nature of state-of-the-art graph- and XAI models. Moreover, the action of the node permutation group challenges the interpretability of latent feature embeddings [8]. In this paper, we generate counterfactual explanations for graph classifiers by utilizing a permutation equivariant graph variational autoencoder to build a semantic latent graph representation. Guided by the classifier, we traverse this latent space to obtain graphs that are semantically similar to the input graph but whose prediction has been altered in a predetermined way. This allows us to answer questions such as "How do we most easily alter a given chemical molecule to improve a specific chemical property?" or "How should a given social network change to reduce its fraud risk score?". We evaluate our approach on three commonly used datasets of molecular graphs.

### 1.1 Background

Counterfactual explanations for graph classifiers take as input a graph and aim to return a minimally altered version of the graph whose class label has been altered. In other words, counterfactual explanations answer the question: "What is the easiest way to change the graph $G$ to alter its classification". For these explanations to be maximally interpretable, in the sense that the factual and counterfactual graph need to be aligned, the pipeline should be permutation equivariant: If the ordering of the input graph nodes is changed, we would like the output graph nodes to be altered accordingly. In this way, any natural alignment between the input and output graphs is preserved.

**Counterfactual Explanations** highlight essential, yet in-sample, changes in input features that affect the outcome of a predictive model, offering valuable insights into the model's decisions. Several approaches exist for tabular [9, 10] and image data [11], as well as graph data [12, 13]. Early works introduce and address key aspects such as sparsity [14, 15], actionability [16, 17], diversity [18, 19] and causality [20, 21]. Recent research focuses on maintaining counterfactuals close to the data manifold, which is usually approximated with generative models such as autoencoders [22–24], flows [25, 26], and diffusion models [27–29]. In the context of graphs, both model-level [30, 31] and instance-level [32, 33] counterfactual explanations have been proposed. Generating counterfactual explanations based on a VAE

---

*Corresponding Author.

has previously been proposed by [33], where a counterfactual prediction loss enables the decoder to generate counterfactual explanations. In this work, we construct instance-level explanations using a permutation equivariant graph variational autoencoder by traversing the latent space appropriately, and *without* explicitly introducing a loss for counterfactual generation during training. Our work, combining the power of a graph generative model and a semantic latent space to generate counterfactual explanations, is to the best of our knowledge, the first to have explicitly trained a graph generative model to do so.

**Equivariant graph generative models.** Equivariance (see Section 2.2 for a rigorous definition) under permutation is especially important in generative and graph-valued prediction tasks. However, many existing graph generative models are not equivariant, as either the encoder is only invariant, or the embedding on the latent space is performed at a graph level and not at a population level [34–36].

To ensure global equivariance, we utilize the permutation equivariant layers of Maron et al. [37] and Pan and Kondor [38]. A permutation equivariant layer can be parameterized using a fixed number of basis elements [39], the number of which are independent of the size of the graphs. An example model using such layers is Hy and Kondor [40], who propose a multiresolution graph variational autoencoder, which is end-to-end permutation equivariant but also rather heavy. We, therefore, choose a simple yet effective PEGVAE [8] with equivariant linear layers [39], overcoming the problems associated with the invariant encoder in alternative VAEs and thus producing an efficient equivariant model.

## 2 Method

We design XAI counterfactual graphs using a variational autoencoder (VAE) to generate in-distribution counterfactual graphs. The generative procedure is built on the assumption that a meaningful latent graph representation has been obtained. This representation space can then be traversed by using the loss of the graph classifier to steer the new graphs toward a class of interest.

Below, we describe the individual components used to build our counterfactual generators, before joining them together in an equivariant framework for counterfactual graph explanation. As graph representations are affected by node ordering, our pipeline is designed to be equivariant with respect to node permutations by using a PEGVAE, and we consider classifiers that are permutation invariant. Fig. 1 shows the complete counterfactual pipeline.

### 2.1 Graph Representation

In the following, a graph $G = (\mathbf{B}, \mathbf{V}, \mathbf{A}, \mathbf{E}) \in \mathcal{G}$ is represented using the following dense matrices:

- $\mathbf{B} \in \{0, 1\}^{n \times 2}$ is a boolean matrix indicating the existence of nodes.
- $\mathbf{V} \in \mathbb{R}^{n \times d_V}$ is a real-valued matrix that contains the node attributes.
- $\mathbf{A} \in \{0, 1\}^{n \times n}$ is a Boolean matrix indicating the existence of edges, the adjacency matrix.
- $\mathbf{E} \in \mathbb{R}^{n \times n \times d_E}$ is a real-valued matrix that contains the edge features if they exist.

Note that in this representation, $n$ is chosen to be the same for all graphs, and graphs containing less than $n$ nodes are zero-padded. This ensures that batching of graphs can be done during model training even if a dense graph representation is employed. Furthermore, while our experiments only consider categorical node and edge attributes, the framework extends to continuous attributes.

### 2.2 Invariance and Equivariance to Permutation

Having defined the graph representation, we define how the permutation group $S_n$ of order $n$ acts on a graph $G \in \mathcal{G}$. Specifically for a permutation $\sigma \in S_n$ we denote the permutation matrix associated with this element as $\mathbf{P}_\sigma$. In this case, we will simply define the group action as

$$\sigma \cdot G := (\mathbf{P}_\sigma \mathbf{B}, \mathbf{P}_\sigma \mathbf{V}, \mathbf{P}_\sigma \mathbf{A} \mathbf{P}_\sigma^\top, \mathbf{P}_\sigma \mathbf{E} \mathbf{P}_\sigma^\top). \quad (1)$$

That is, the rows and columns of $\mathbf{A}$ and $\mathbf{E}$ are permuted, whereas only the rows of $\mathbf{B}$ and $\mathbf{V}$ are permuted.

It is widely acknowledged that predictive graph models should incorporate the notions of invariance and equivariance [41]. In the case of classification, a graph classifier should not depend on the node ordering. The graph classification model $\mathcal{C} : \mathcal{G} \to \{0, 1\}$ considered in this paper is therefore designed to be invariant, that is, $\mathcal{C}(\sigma \cdot G) = \mathcal{C}(G)$ for any $G \in \mathcal{G}$ and $\sigma \in S_n$.

Likewise, we design the autoencoder to be equivariant to permutation. Thus, the encoder $\mathcal{F} : \mathcal{G} \to \mathbb{R}^n$ and decoder $\mathcal{D} : \mathbb{R}^n \to \mathcal{G}$ should have the property, that for any $G \in \mathcal{G}$:

$$\mathcal{D}(\mathcal{F}(\sigma \cdot G)) = \sigma \cdot \bar{G} \quad \text{for all } \sigma \in S_n. \quad (2)$$

Where $\bar{G}$ denotes the reconstruction of $G$. For graph autoencoders, equivariance is crucial: We need input and output graphs to be aligned when computing the loss during training. This also applies to assessing counterfactual graph explanations.

Several ways of creating equivariant layers can be considered, but for the networks in this paper, we consider a specific basic building block consisting

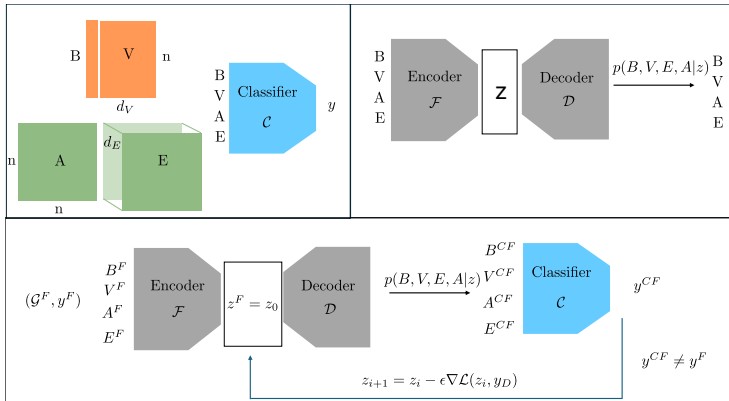

**Figure 1.** Top: The classifier architecture and the PEGVAE. Bottom: The counterfactual graph generation.

of: A linear equivariant layer as specified by Maron et al. [39] and a non-linearity consisting of node- and edge-wise aggregations implemented as convolutions with a kernel size of one and the ReLU activation function. Batch normalization is applied before the activation. We will refer to this construction as an equivariant/invariant *module* depending on the applied equivariant/invariant linear layer. The linear equivariant layers are constructed using the fact, that any permutation equivariant linear function $L : \mathbb{R}^{n^k \times d} \to \mathbb{R}^{n^l \times d'}$ can be expressed using exactly $b(k + l)dd'$ known basis elements, where $b(\cdot)$ represents the Bell number, i.e. the number of possible partitions of a set of a certain size. Based on this result, we can define equivariant linear layers at the node and edge levels by forming a weighted linear combination of the known basis elements. In the case where $k = l = 2$, this amounts to $b(4)dd' = 15dd'$ trainable weights. See for Hansen et al. [8], Maron et al. [37], Pan and Kondor [38], and Thiede et al. [42] for further uses of this type of layer.

## 2.3 PEGVAE

We design a PEGVAE similar to Hansen et al. [8]. We employ a standard Gaussian prior on the latent space variable, i.e. $p(\mathbf{z}) = \mathcal{N}(\mathbf{z} \mid \mathbf{0}, \mathbf{1})$, and let $q_\phi(\mathbf{z} \mid G)$ denote the approximate posterior also being Gaussian, and where $\phi$ refers to the parameters of the encoder as is custom when dealing with VAEs. Specifically, we factorize the likelihood $p_\theta(G \mid \mathbf{z})$ as:

$$p_\theta(G \mid \mathbf{z}) = p_{\theta_B}(\mathbf{B} \mid \mathbf{z})p_{\theta_V}(\mathbf{V} \mid \mathbf{z}, \mathbf{B})$$
$$p_{\theta_A}(\mathbf{A} \mid \mathbf{z}, \mathbf{B}, \mathbf{V}) \qquad (3)$$
$$p_{\theta_E}(\mathbf{E} \mid \mathbf{z}, \mathbf{B}, \mathbf{V}, \mathbf{A}),$$

where $\theta = \{\theta_B, \theta_V, \theta_A, \theta_E\}$ refers to the parameters of the decoder. Thus, when sampling a new graph from $p(G)$, we first sample the prior $p(\mathbf{z})$, then $p(\mathbf{B} \mid \mathbf{z})$, etc., until a graph has been obtained. This factorization makes it easier to sample valid graphs than if $p(G \mid \mathbf{z})$ were modeled directly: We can constrain edges to be sampled between nodes that

exist, and assign classes only to nodes and edges that exist. We now minimize the negative evidence lower bound (ELBO), which is given by, i.e.:

$$-ELBO = \mathbb{E}_{\mathbf{z} \sim q_\phi(\mathbf{z}|G)}[-\log p_\theta(G \mid \mathbf{z})]$$
$$+ KL(q_\phi(\mathbf{z} \mid G) \parallel p(\mathbf{z})), \qquad (4)$$

where $KL(\cdot \parallel \cdot)$ denotes the Kullback-Leibler divergence. Here, the negative log-likelihood works as a *reconstruction loss*, and the KL-loss works as regularization. Since the prior is Gaussian with mean zero, and thus has a symmetric density, the node ordering will not affect the size of this term. In practice, we tune the KL-loss with a hyperparameter $\beta \in [0, 1]$ [43] by optimizing,

$$\mathbb{E}_{\mathbf{z} \sim q_\phi(\mathbf{z}|G)}[-\log p_\theta(G \mid \mathbf{z})] + \beta \cdot KL(q_\phi(\mathbf{z} \mid G) \parallel p(\mathbf{z})).$$

The exact architecture of the permutation equivariant VAE as well as any hyperparameters used during training can be found in Appendix A.1.2.

## 2.4 Classifier Design

As the PEGVAE is permutation equivariant, the graph classifier $\mathcal{C} : \mathcal{G} \to \{0, 1\}$ is designed to be permutation *invariant* as the prediction should not be affected by node ordering. Following Bronstein et al. [41], this feature is obtained by composing a number of equivariant layers followed by an invariant global pooling layer. The output of this global pooling layer can be viewed as a graph embedding, which is passed to a fully connected neural network to obtain probabilistic class predictions. The architecture of the classifier, as well as the hyperparameters used during training, can be found in Appendix A.1.1.

## 2.5 Generating Counterfactuals via Latent Space Traversal

Given a factual graph $G^F$ we generate counterfactual explanations for the class prediction. First, a latent encoding $z^F$ of $G^F$ is found by evaluating $\mathcal{F}(G^F)$.

**Algorithm 1:** CGCF

**Input:** $G^F$ and $N \geq 0$
**Output:** $G^{CF}$
$\mathbf{z}_0 \leftarrow \mathcal{F}(G^F)$;
$y_D \leftarrow 1 - y_F$;
$i \leftarrow 1$;
**while** $i \leq N$ **do**
    $G_i^{CF} \leftarrow \mathcal{D}(\mathbf{z}_i)$;
    $y_i \leftarrow \mathcal{C}(G_i^{CF})$;
    **if** $y_i = y_D$ **then**
        $\text{mask} \leftarrow 0$;
    **else**
        $\text{mask} \leftarrow 1$;
    **end**
    $\mathbf{z}_{i+1} \leftarrow \mathbf{z}_i - \epsilon \cdot \nabla \mathcal{L}(\mathbf{z}_i, y_D) \cdot \mathbf{mask}$;
    $i \leftarrow i + 1$;
**end**
$G^{CF} \leftarrow G_N^{CF}$;
**return** $G^{CF}$

Secondly, a counterfactual graph $G^{CF}$ is generated by iterative updates using gradient descent with a learning rate of $\epsilon > 0$, defined as:

$$\mathbf{z}_{i+1} = \mathbf{z}_i - \epsilon \nabla \mathcal{L}(\mathbf{z}_i, y_D), \tag{5}$$

where $\mathbf{z}_0 = \mathbf{z}^F$, and $\mathcal{D}$ denotes the decoder, $\mathcal{C}$ denotes the classifier, and $y_D$ denotes the *desired* label of the counterfactual. The loss function $\mathcal{L}$ is defined as a cross entropy loss with L2-regularization limiting the size of $z_i$, i.e.:

$$\mathcal{L}(\mathbf{z}_i, y_D) = -\sum_{k=1}^{K} \mathbb{1}_{\{D=k\}}(y_D) \log(y_i) + \lambda \|\mathbf{z}_i\| \tag{6}$$

for $\lambda \geq 0$, and where $y_i = (\mathcal{C} \circ \mathcal{D})(\mathbf{z}_i)$. The updates stop when either $y_i = y_D$, or when the maximum number of iterations, which is set as a hyperparameter, has been reached. This process results in an estimated latent representation $\mathbf{z}^{CF}$ with associated counterfactual graph $G^{CF} = \mathcal{D}(\mathbf{z}^{CF})$ and associated class prediction $y^{CF} = \mathcal{C}(G^{CF})$. The counterfactual graph is inferred from the distribution produced by the decoder as described below.

**Inferring Graph Reconstruction.** A key part of our pipeline for generating counterfactual explanations is to iteratively update a latent code $\mathbf{z}$ by passing it through a pipeline consisting of the pre-trained decoder and a pre-trained, potentially unknown classifier for each iteration. However, the *decoder* is parameterizing $p_\theta(G \mid \mathbf{z})$, and thus the output of the decoder is a distribution and *not* in the graph domain. Sampling from $p_\theta(G \mid \mathbf{z})$ would be a solution, but would not allow backpropagation. To alleviate this problem we use a Gumbel-Softmax distribution to approximate $p_\theta(G \mid \mathbf{z})$ as outlined

by Jang et al. [44], enabling us to compute gradients using a reparametrization trick.

Alternative approaches include sampling multiple graphs from the likelihood passing each one through the decoder, computing the loss of the classifier with respect to the average prediction loss, or picking the graph that maximizes the likelihood $p_\theta(G \mid \mathbf{z})$.

**Algorithmic Representation.** Alg. 1 describes how updates are iteratively made to generate counterfactuals using CGCF. In general, the procedure follows the description from Section 2.5. However, from the algorithm, one can clearly see how updates can be made to batches of counterfactuals by using a mask. This ensures that after the desired label has been achieved, then no more updates will be done to the latent code. Also note, that to ensure that the algorithm does terminate it runs for $N \in \mathbb{N}$ steps, and not until the classifier assigns the desired label to the generated graph. As a consequence of this the procedure does not guarantee that a counterfactual is obtained.

# 3 Experiments

## 3.1 Data

We evaluate our method using three molecular graph datasets: NCI1 [45], Mutagenicity [46–48] and AIDS [47, 49], where graph nodes represent atoms and edges represent bonds. Each dataset poses a binary classification task: Whether the molecule is active against HIV or not (AIDS), whether it is mutagenic or not (Mutagenicity), and whether the molecule is anticancer (NCI1). We follow the graph pre-processing and filtering of Huang et al. [50], including only molecules with nodes occurring with a frequency larger than 50 in the dataset to avoid imbalance. Furthermore, as our architecture relies on a dense graph representation, we only consider graphs with fewer than 50 nodes (Mutagenicity, NCI1) and 30 nodes (AIDS). Each dataset is divided into training, validation, and test sets, with 10% allocated to test and validation. Additional information can be found in Tab. A.1 in Appendix A.

## 3.2 Evaluation Metrics

The counterfactuals quality of the counterfactuals are evaluated as a trade-off between *identity preservation*, i.e, the degree to which the generated counterfactual graph resembles the factual graph, and *validity*, i.e, whether the generated counterfactual is indeed a valid candidate for the desired class.

**Identity Preservation.** Three identity preservation measures are used. *Graph Edit Distance* (GED), measures the graph distance between factual and

**Table 1.** Validation Results

| | | GED ↓ | LED ↓ | Cosine Similarity ↑ | SIC ↑ | Flip-Ratio ↑ |
|---|---|---|---|---|---|---|
| Aids | Random | $63.4 \pm 19.63$ | $14.95 \pm 2.62$ | $-0.01 \pm 0.41$ | $0.83 \pm 0.38$ | $0.83$ |
| | Graph of NN from Training | $\mathbf{18.54 \pm 9.95}$ | $8.31 \pm 1.82$ | $\mathbf{0.15 \pm 0.38}$ | $0.4 \pm 0.43$ | $0.33$ |
| | Decoded Mean of k-NN | $47.66 \pm 10.84$ | $8.44 \pm 1.95$ | $-0.19 \pm 0.14$ | $\mathbf{0.99 \pm 0.03}$ | $\mathbf{1.0}$ |
| | Classifier Guided CF | $35.79 \pm 14.01$ | $\mathbf{7.11 \pm 2.35}$ | $-0.08 \pm 0.25$ | $0.94 \pm 0.15$ | $0.98$ |
| Mutagenicity | Random | $65.28 \pm 19.84$ | $8.53 \pm 0.84$ | $0.03 \pm 0.32$ | $0.16 \pm 0.37$ | $0.52$ |
| | Graph of NN from Training | $\mathbf{28.1 \pm 17.87}$ | $1.24 \pm 0.45$ | $\mathbf{0.59 \pm 0.34}$ | $0.14 \pm 0.25$ | $0.44$ |
| | Decoded Mean of k-NN | $41.24 \pm 18.93$ | $\mathbf{1.05 \pm 0.36}$ | $0.42 \pm 0.39$ | $0.09 \pm 0.3$ | $0.41$ |
| | Classifier Guided CF | $50.62 \pm 21.41$ | $3.63 \pm 2.09$ | $0.1 \pm 0.46$ | $\mathbf{0.42 \pm 0.33}$ | $\mathbf{0.92}$ |
| NCI1 | Random | $59.94 \pm 17.01$ | $41.04 \pm 3.18$ | $-0.01 \pm 0.21$ | $0.11 \pm 0.43$ | $0.49$ |
| | Graph of NN from Training | $\mathbf{40.51 \pm 16.44}$ | $8.67 \pm 3.29$ | $\mathbf{0.58 \pm 0.35}$ | $0.15 \pm 0.25$ | $0.58$ |
| | Decoded Mean of k-NN | $42.59 \pm 15.82$ | $\mathbf{8.0 \pm 3.04}$ | $0.38 \pm 0.3$ | $0.13 \pm 0.34$ | $0.51$ |
| | Classifier Guided CF | $53.14 \pm 23.01$ | $13.19 \pm 12.1$ | $0.3 \pm 0.44$ | $\mathbf{0.34 \pm 0.25}$ | $\mathbf{1.0}$ |

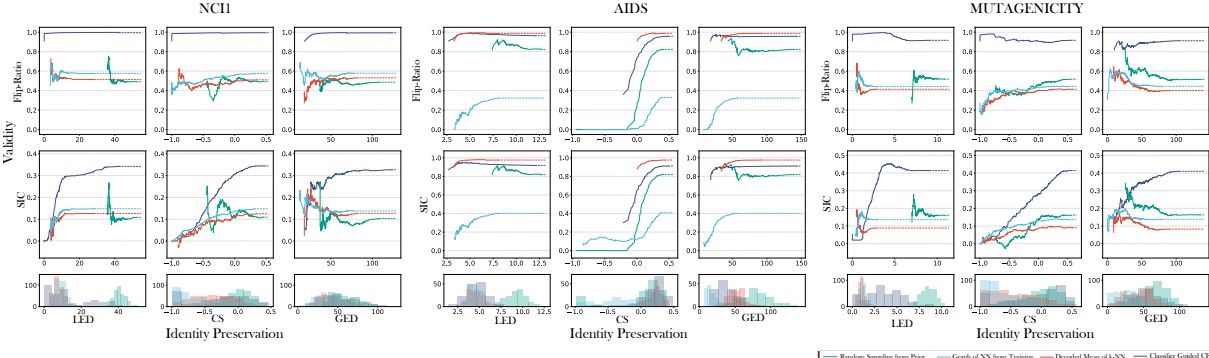

**Figure 2.** Trade-off between metrics for Identity Preservation and Validity.

counterfactual graphs. *Latent Euclidean Distance* (LED) refers to the Euclidean distance between the latent representation of two graphs. *Cosine Similarity* (CS) is computed between the graph embeddings extracted from the classifier.

**Validity.** Two validity measures are used. *Flip-Ratio* (FR) refers to the number proportion of generated counterfactual explanations, which are assigned the desired class by the classifier. On top of that, we measure the *Signed Increase in Confidence* (SIC), referring to the increase or decrease classifier's confidence for the desired class. Specifically, we compute $p_{d_{CF}} - p_{d_F}$, where $p_{d_{CF}}, p_{d_F} \in [0, 1]$ are classifier's output, that the generated factual and counterfactual graphs belong to the desired class.

### 3.3 Baselines

We include several baseline counterfactual methods. First, counterfactual explanations can be generated naively by decoding random samples from $p(\mathbf{z})$. We refer to this method as **Random**. That is, we first sample from the prior $p(\mathbf{z})$ and subsequently from $p_\theta(G \mid \mathbf{z})$. This serves as a naïve baseline, as nothing promotes the generation of a counterfactual.

The **Graph of NN from Training** counterfactual uses a factual latent code $\mathbf{z}_F$ and a desired label $y_D$ to pick the closest latent code of graphs in the training dataset with the desired label. We then let

the original training set graph be the counterfactual explanation. This method is guaranteed to generate a valid graph of the desired class. However, the classifier may not classify this correctly, and thus the explanatory power of these graphs may be limited.

A more realistic counterfactual method is given by **Decoded Mean of k-NN**. Here, given a factual latent code $\mathbf{z}_F$ and a desired label $y_D$, we pick the $k$ closest latent codes of graphs in the training dataset with the desired label. These latent codes are then averaged and decoded using the VAE. For this work, we pick $k = 10$. We expect this method to generate counterfactual explanations well within the decision boundary of the classifier.

As this work is preliminary, and since we investigate whether it is viable to generate counterfactual explanations based on traversal of a latent space of a VAE without having to explicitly train it for generating counterfactual explanations, all the methods used for comparison are based on the generating counterfactuals from the latent space of the same VAE. We will refer to the method outlined in Section 2.5 as **Classifier Guided Counterfactuals (CGCF)**. Source code for running each baseline will be publicly available at https://github.com/abildtrup/latent-graph-counterfactuals.

## 3.4 Results

Tab. 1 shows a variety of identity preservation and validity metrics for counterfactual quality, computed on the test set. A good counterfactual has high validity, while still maintaining a high degree of identity preservation of the original input. Note that the classifier-guided counterfactuals consistently outperform the baselines measured via flip-ratio, only occasionally surpassed by the Decoded Mean of k-NN on AIDS. Unsurprisingly, the best-performing method in terms of identity preservation is often just the Graph of NN from Training method. However, we also see, that this method consistently performs poorly in terms of validity, underscoring the fact that a trade-off is present, and the evaluation of the method based on the table alone is not sufficient.

Fig. 2 depicts the trade-off between identity preservation (columns) and validity (rows) for each dataset. The curve is constructed by computing the average validity score from all samples with an identity preservation score below a certain threshold, given by the x-axis. Note that we only compute the average after having made 10 observations with the validity below the threshold given by the x-axis before we compute an average to remove noise. In the case of the Flip-Ratio this can be interpreted as an estimate for the probability of the counterfactual graph having successfully flipped the class of its factual graph, given that the level of identity preservation is below a certain threshold. We observe that the classifier-guided counterfactuals achieve tradeoffs competitive with all baselines, and outperform on NCI1 and Mutagenicity. One should note, that to make the interpretation of the plots the same (high values being desirable, and low values being undesireable) we consider the *negative* cosine similarity. The histograms of Fig. 2 show the distribution of identity preservation scores of the generated counterfactuals for each score.

## 4 Discussion

**Classifier Guided CF is robust on all datasets.**
On AIDS, the Decoded Mean of k-NN and CGCF achieve good validity scores, as well as good tradeoffs between identity preservation and validity. However, only the CGCF proves to be robust across all datasets considered here. It is worth noting that this may be because the AIDS dataset contains graphs with fewer nodes (30) than Mutagenicity (50) and NCI1 (50). As the dimensionality of our VAE latent space matches the number of nodes, this could explain the decrease in performance. Also note that even though the Graph of NN from Training method is guaranteed to generate a counterfactual of the desired class, the validity for this method is in general very low, as it will often generate a graph that will be misclassified by the classifier. See Appendix A.3 for an illustration of this phenomenon.

**Generating counterfactual explanations based on latent space exploration removes the need for defining graph distance explicitly.** When producing counterfactual explanations, we generally want to find a counterfactual graph

$$G^{CF} = \arg \min_G \text{dist}(G, G^F)$$

such that $\mathcal{C}(G^F) \neq \mathcal{C}(G) = y_D$ for some notion of graph distance $d$ and some desired label $y_D$. This ensures that the counterfactual graph is *similar* to the original (factual) graph. However, the choice of graph distance will always be open and likely depends heavily on the application at hand. When counterfactual graphs are generated based on latent space traversal, we do not have to define an explicit metric on the graph space. Our method relies on the distance between latent graph representations, making it widely usable across different cases.

**The method enables the construction of an arbitrary number of possible explanations.** Our method is probabilistic and models the likelihood $p_\theta(G \mid \mathbf{z})$. Thus, we can easily sample an unlimited amount of counterfactual explanations given some factual graphs.

**Limitations.** Generating counterfactuals based on a pre-trained VAE means that the quality of the method relies heavily on the quality of the generative model. For instance, if a latent counterfactual is obtained on the classification boundary in the latent space far away from any latent training point, then the sampled counterfactual may not be reasonable. In this work, we focus on exploring instance-level explanations. Extending our method to model-level explanations by exploring general patterns that arise when moving toward the counterfactual class will be a subject of future work.

## 5 Conclusion

We proposed to generate counterfactual explanations for graph classification using the latent representation space of a permutation-equivariant VAE. We find it a promising direction of research to utilize a preexisting generative model, such as a VAE, to generate valid counterfactual explanations for graphs without having to define explicit graph metrics. As shown by our experiments, the classifier-guided counterfactual explanations provide a robust trade-off between identity preservation and validity.

# Acknowledgements

This work was supported in part by the Independent Research Fund Denmark (grant no. 1032-00349B), by the Novo Nordisk Foundation through the Center for Basic Machine Learning Research in Life Science (grant no. NNF20OC0062606), the Pioneer Centre for AI, DNRF grant nr P1, the DIREC project EXPLAIN-ME (9142-00001B), and by the ERC Advanced grant 786854 on Geometric Statistics.

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

# A  Appendix

## A.1  Evaluation of auxiliary models.

The success of our suggested method for generating counterfactual explanations is highly dependent on the quality of the models considered. The classifier and the VAE considered in this work are both designed to be permutation invariant and equivariant respectively.

### A.1.1  Classifier

The classifier used for all datasets consists of four equivariant modules (as described in Section 2.2) producing node embedding. These are then follows be an invariant max-pooling operation ensuring the model as a whole is invariant. The embeddings obtained after this operation are the graph-embeddings. Each module has 20 channels. These graph-embeddings are then passed through a fully connected neural network with 200 neurons. Note, that the first module only considers the $\mathbf{B}$ and $V$ matrices, and the output is appended to the $A$ and $E$ matrices for subsequent processing. The classifier was trained for 100 epochs using a learning rate of 0.001, and a batch-size of 64. In Tab. A.2 the AUROC computed on the test-set is reported for all classifiers.

### A.1.2  Permutation equivariant VAE

The permutation equivariant VAE is designed using modules similar to the ones used in the classifier. The encoder consists of 4 equivariant modules (as described in Section 2.2) each of which deals with 20 channels. Again, the first module only considers the $\mathbf{B}$ and $\mathbf{V}$ matrices. The last layer is divided into two parts which outputs the mean and the log variance of the approximate posterior respectively. The decoder is comprised of four separate decoders; one for $\mathbf{B}, \mathbf{V}, \mathbf{A}$ and $\mathbf{E}$ respectively. The decoders $\mathbf{B}$ and $\mathbf{V}$ consists of one equivariant module each, where the decoders for $\mathbf{A}$ and $\mathbf{E}$ consists of two modules. In Tab. A.2 the losses relevant for the VAE are reported. All values are computed on the test-set.

For the PEGVAE trained on the Aids dataset a learning rate of 0.001 was used, along a learning rate scheduler configured to halfing the learning rate if the validation loss had not improved for 150 epochs. In total training ran for 2000 epochs. For the trade-off between the KL-loss and the reconstruction loss $\beta = 0.1$ was chosen. Additionally, to avoid posterior collapse, the $\beta$ was increased gradually during a burn-in period. For the NCI1 and Mutagenicity datasets a similar training setup was employed with the sole change that we set $\beta = 0.5$.

**Table A.1.** Dataset statistics after pre-processing.

| Dataset | #Graphs | Maximum #Nodes | #Node Attributes | #Edge Attributes |
|---|---|---|---|---|
| AIDS | 1635 | 30 | 9 | 3 |
| MUTAGENICITY | 3935 | 50 | 10 | 3 |
| NCI1 | 3678 | 50 | 10 | ✗ |

**Table A.2.** Performance of auxiliary models.

| | KL - PEGVAE | Reconstruction Error - PEGVAE | ELBO - PEGVAE | AUROC - Classifier |
|---|---|---|---|---|
| AIDS | 31.21 | 48.04 | 79.91 | 0.99 |
| Mutagenicity | 40.82 | 166.52 | 207.35 | 0.82 |
| NCI1 | 34.19 | 192.88 | 226.48 | 0.79 |

as the nearest neighbor for several of the triangles, which in turn will also be misclassified. As such a single misclassified graph can have a disproportionate impact on the validity score. Note that this is a thought up example and serves only to illustrate the low performance on validity for this method.

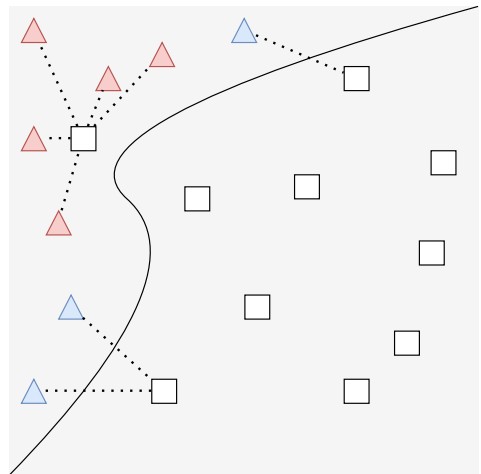

**Figure A.1.** Illustration of how generating counterfactual explanations based on the nearest neighbor can produce low validity.

## A.2 Hyperparameter Selection for Genration of Counterfactuals

For the Classifier Guided CF method optimization was done using an Adam optimizer with a learning rate of 0.05 for 1 000 iterations and $\lambda = 1$ used for regularization. The hyperparameter $\tau$ used for the Gumbel-Softmax was also set to 1. This parameter determines how close an approximation the Gumbel-Softmax is to the desired discrete, categorical distribution as opposed to a uniform distribution assigning equal probability mass to all classes. For the Decoded Mean of k-NN method, we choose $k = 10$.

## A.3 Illustration of Nearest Neighbor Graph Counterfactual with Low Validity

In figure A.1 we illustrate intuitively how producing counterfactual explanations based on the Graph of NN from Training method can produce low validity scores even though, the counterfactuals produced are guaranteed to be from the opposite class. In the depicted examples graphs are considered divided into two classes: Triangles and squares, and these two classes are almost perfectly separated by the classification boundary, with only one square being classified as a triangle. However, since the representation which we have obtained through the VAE to a large degree clusters latent codes of the same type, this single misclassified graph will be chosen

