# OpenReview forum: "Graph Counterfactual Explainable AI via Latent Space Traversal"
_NLDL.org/2025/Conference — NLDL 2025 Oral_

### Official Review · Reviewer_5Zk4 · 2024-10-07
**preliminary but sensible contribution**

**Confidence:** 4

**Summary:**

The paper introduces a pipeline for generating (in-distribution) counterfactual
explanations of GNN decisions.  An equivariant generative graph model is used
for the purpose.  Specifically, the idea is to model a generative distribution
over in-distribution graphs as a VAE;  the output of this VAE is fed to the
classifier;  a counterfactual is computed by traversing the latent space (via
gradient ascent/descent) in a direction that crosses the decision boundary.
An empirical evaluation is carried out against three simple but reasonable
baselines on three data sets.

**Strengths:**

- Generally well written and structured.

- Tackles a challenging problem: generating counterfactuals for
  relational-continuous data.

- The proposed method is intuitively sensible.

- Experiments are relatively basic but set up appropriately.

- I appreciated that the authors do point out limitations of their proposed method.

**Weaknesses:**

- There already exist works for generating counterfactuals that rely on
generative models in the context of non-relational data.  It would make sense to
mention a couple, and to explain that for graphs the problem is more challenging
due to equivariance etc.

- Section 2.1: it would have helped to provide a basic intuition behind the
layers by Maron et al.

- The notion of "fidelity" (aka "faithfulness" in the literature) normally
  refers to the degree by which an explanation is faithful to the reasoning
  process of a model, see for instance [1].  I would refrain from using the same
  name for a different quantity: it is confusing.

- The choice of competitors is limited and the experiments are run using a single
  GNN model; hinting at the fact that this is an
  (understandably) preliminary work.

- In a similar vein, it would make sense to discuss the direct competitors more
  in details and compare (both conceptually and empirically) against them.


**Suggestions**:

- Please name your method.  It makes it easier to reference it in future work.
  You can also use it in the plots to make it easier to identify your proposed
  approach (the dark blue bar).

- Using $\mathcal{C}$ for the GNN classifier and $\mathcal{F}-\mathcal{D}$ for
the encoder-decoder is a bit odd - calligraphic math symbols are normally used
for denoting sets (as you do for the set of graphs).

- For conditional distribution, consider using $\mid$ instead of $|$.

- Figure 1 should probably be in page 2 (because it helps to figure out what
  latent space is being traversed.)

[1] Agarwal, C., Zitnik, M., & Lakkaraju, H. (2022, May). Probing gnn explainers: A rigorous theoretical and empirical analysis of gnn explanation methods. In International Conference on Artificial Intelligence and Statistics (pp. 8969-8996). PMLR.

- The description of the classifier should probably be an experimental detail,
  while it currently resides in the Method section.

- What is the reason for reporting the distance in latent space?  Does it matter
for practical purposes?

**Final Rebuttal Confidence:**

4

**Final Rebuttal Justification:**

I was quite positive about the paper to begin with, and the rebuttal has managed to clarify the few issues I had pointed out.

**Justification:**

All in all, a preliminary but promising contribution.  The proposed approach is sensible and the (preliminary) evaluation shows promise.  The paper is currently missing a discussion of advantages/disadvantages wrt existing works, as well as a thorough empirical comparison.

---

> ### Author Rebuttal · Authors · 2024-10-22
>
> Thank you for reading our paper, and providing feedback. We are glad that the paper is well received overall. We will update the layout of the paper appropriately, and try to find a catchy name. The changes will be uploaded in a revised version of the paper before the rebuttal period ends. The rest of your concerns we will address below.
>
> **Additional related work:** Indeed, the section on counterfactual explanations can be extended further to include citations to counterfactual explanations for non-relational data. Due to restrictions on space in the early submission phase we had to be very concise, but we will provide a couple of references and discussion.
>
> **Basic intuition of equivariant layers:** Thank you for the input. We would like to accommodate the request, but due to space constraints we will add a section in the appendix describing the overall construction of the layers.
>
> **Notion of fidelity:** We will address the ambiguity of using the term “fidelity” in this context. It will be substituted for “Identity Preservation” in the final paper.
>
> **Limited choice of competitors:** As you mention our work is preliminary in nature, and we investigate whether it is viable to generate counterfactual explanations based on traversal of a latent space of a VAE without having to explicitly train it for generating counterfactual explanations. As such the methods we compare are all based on the generating counterfactuals from the latent space of the same VAE. We do not claim that it is state-of-the-art, but we do think our results show that the approach is promising, a sensible way of approaching the problem.
>
> **Latent space distance:** As we address in the discussion, we aim to find the “closest” graph for which the classifier assigns the desired counterfactual class. However, when dealing with complex data objects, the “notion” of closeness to pick is not clear. As such we evaluate several metrics (i.e. the fidelity metrics) to provide a more clear picture. In Fig 2, we can then evaluate the consistency between the considered metrics. Including the Euclidean distance in the latent space is based on the assumption that “similar” graphs will be close in the latent space of a well trained VAE. As such using pairwise distances in the latent space offers a way to not having to define distances in the dataspace, in our case distances between molecules, which would be very heuristic in nature.

---

### Official Review · Reviewer_DH66 · 2024-10-07
**This paper presents several intriguing ideas but my current assessment is just below the acceptance threshold.**

**Confidence:** 5

**Summary:**

This paper presents several intriguing ideas. Firstly, it introduces a generative modeling method called PEGVAE, which utilizes VAE for graph instances. Unlike image instances, graph instances require a guarantee of permutation invariance during the learning process. This aspect, which has not been considered in traditional image-based VAEs, is likely to engage readers' interest. The authors effectively detail this novel framework in sections 1.Introduction and 2.1-2.2.

Secondly, the paper proposes an innovative approach for iteratively identifying counterfactual latent representations by utilizing gradients derived from a classifier when input graphs are classified. The quality of the discovered counterfactuals is evaluated using two explainability principles: fidelity and validity.

**Strengths:**

Overall, the paper is comprehensive, and the key points are well articulated. If the authors could address the following suggestions during the review process and incorporate them into the camera-ready version, I would be open to reconsidering my current assessment, which is just below the acceptance threshold.

**Weaknesses:**

Points to improve:

1. Syntax errors and typos: The manuscript contains numerous syntax errors and typographical mistakes. I urge the authors to thoroughly proofread the entire document to improve clarity. Some errors I noted include:
L175: \rightarrow should be \leftarrow
L197: Appendix B is missing
L199: N(0|1) should be N(0, 1)
L204: Eq. (3): p(E|B,V,E) should be p(E|B,V,A)

2. Algorithmic representation of the process: Including an algorithmic representation of the process would greatly enhance readers' understanding. Currently, the method by which the PEGVAE-classifier is trained (whether end-to-end or separately) and how counterfactual samples are identified for specific instances is not clearly delineated. Additionally, the batching of graphs should be included in the algorithmic description.

3. Discussion section: The discussion is somewhat lacking. It would be beneficial to explore whether graph instances belonging to the same class exhibit similar counterfactuals. Moreover, if latent features are altered using gradients obtained from intentionally misclassified instances within a pre-trained PEGVAE-Classifier setup, what would the reconstructed graphs look like? Addressing these questions would add depth to the analysis.

4. Quality of counterfactual explanations: On L406, the statement that "generative models do not guarantee quality counterfactual explanations" raises a question about whether PEGVAE falls into this category as well. This remark could diminish the persuasive power of the paper. Instead, I recommend visualizing the generated graph samples and including a more detailed examination and analysis of the contexts in which they are produced. If further experiments are conducted in the meantime, it would be beneficial to include those results during this review process.

**Final Rebuttal Confidence:**

5

**Final Rebuttal Justification:**

Thank you for your efforts in addressing my questions. Your response is comprehensive and I would like to change my decision to accept this paper for NLDL 2025. Please ensure that the answer regarding the first inquiry, which relates to point 2 in my initial comments, is included in the camera-ready version.

**Justification:**

Although this paper attempts to address an important permutation invariance problem through an innovative approach, the experimental results do not align with the intended outcomes, leading me to recommend against acceptance.

---

> ### Author Rebuttal · Authors · 2024-10-22
>
> Thank you for your review. We are happy that your overall impression of the paper is positive. In the following we will address your concerns to the best of our ability, and we will upload a new version of the paper before the end of the rebuttal period using the suggestions of all reviewers.
>
> **Syntax errors and algorithmic:** Thank you for pointing this out, we will update our manuscript regarding syntax errors and the algorithmic representation as you suggest.
>
> **Additional Weaknesses:** You raise some interesting points on additional experiments which could be conducted for the evaluation of counterfactuals.
>
> Your second inquiry is regarding how the latent codes are updated for graphs that are misclassified, i.e. the classifier will not be able to identify the factual class. Now, if the reconstruction of the factual graph is also misclassified, it means that it is assigned the desired class of the counterfactual, and as such, the algorithm will stop, and no updates will be made to the latent code.
>
> The first inquiry is regarding whether graphs belonging to the same class would exhibit similar counterfactuals. We are not sure exactly what you mean, but we try to elaborate. First of all, since the counterfactuals stay close to the original graphs, they do not all look the same – they preserve some of the variation from the original graphs. However, if you are looking for more general patterns that arise when moving towards the counterfactual class, this resembles model level counterfactual explanations. However, the method we currently propose is for instance level counterfactual explanations. From our perspective, extending this method by analyzing the similarity of the generated counterfactuals would likely require substantial problem specific engineering, which, we fear, would generalize poorly to other problems. As such we do not include it in the current paper, and limit the scope to showing how the generating graph counterfactuals based on the latent space of a VAE without providing a counterfactual loss as a training target is indeed possible.
>
> We would be happy to include all of the above points in our discussion. If there is anything which we have misunderstood or left out, then please let us know.
>
> Overall thank you for your input. We will take your suggestions, as well as the ones given by the rest of the reviewers, to ensure that the paper is persuasive and clear.

---

### Official Review · Reviewer_mmEe · 2024-10-07
**Clear approach to counterfactual explanations for graph classifiers**

**Confidence:** 4

**Summary:**

This paper presents a method that produces counterfactual explanations for graph classifiers. At the core of the method, there is a VAE, which allows the authors to avoid the tricky problem of defining a distance on the space of graphs by moving into VAE's latent space instead.

**Strengths:**

**Coherence and clarity**

The paper is well structured, starting with a clear description of the problem and the related difficulties, then proceeding to explain the method, and finally commenting on the experimental results. I will comment on some elements that affect the clarity of some parts of the paper among the weaknesses, but overall the paper is clear.


**Incremental contribution**

The authors clearly state what's new in their work, namely the use of a permutation equivariant GVAE. I'm not super familiar with the specific topic, but a quick search in the literature seems to confirm that this is, indeed, a new element worth of being published and discussed.


**Questions**

I have three questions for the authors:
1. I was wondering how robust is your method against different types of classifiers and how sensitive it is to the reconstruction power of the GVAE.
2. You only considered binary classification tasks. Would your approach work also for more classes?
3. Mutagenicity dataset comes with ground truth, is it possible to use those to measure the performance of your method?

**Weaknesses:**

**Clarity**:
- I would recommend to the authors to take the opportunity of this review to fix a few typos. There are not many and they don't affect the overall comprehensibility. I'll make a couple of examples here. Sometimes the latent variable $z$ is written in bold (see eq. 4), sometimes it's not (see section 2.5). Lines 265, 267 and 270 have extra parentheses.
- There are some parts of the paper that could be a bit clearer, maybe you can make use of the extra page to expand and clarify them.
  - In line 046 you talk about "*dataset-specific* permutations equivariant GVAE", but it's not clear to me what makes it dataset-specific.
  - In line 064 you talk about the necessity for explanations to be "maximally interpretable", without specifying what that means.
  - You introduce the difference between equivariance and invariance quite early in the paper (lines 96-98), maybe you could specify that you will explain it better in section 2.2.
  - In line 099 it's not clear to me the difference between graph level and population level embeddings, and how that impacts the equivariance of the representation.
  - As a topological distance, you use the graph edit distance. Any reason why you use this specific one instead of others?
  - It's a bit unclear to me what the Signed Increase in Confidence is. Maybe you could expand it a bit better?
- The readability of table 1 could be improved, for example:
  - by introducing acronyms for the metrics, so that you can increase the font size;
  - by using bold numbers also in the part related to the fidelity to indicate the best-performing methods.
- The readability of figure 2 could be improved, for example:
  - by making them larger;
  - introducing the acronyms (GED is not explicitly defined in the text);
  - "Mean Absolute Difference" is not defined within the text and "Latent Cossimilarity" has a different name in text;
  - maybe you could show only one dataset with larger figures, and move the others in the appendix?
  - how should the histograms be interpreted?
- Since the very beginning of the paper I was expecting to see some factual and counterfactual graphs plotted somewhere. It's good to have numbers and metrics, but maybe one example could help ground the discussion.
- You mention that Nearest Neighbor counterfactuals have low validity because they will be *often* misclassified by the classifier. I understand that it can happen, but why does it happen often?

**Reproducibility**

The method and the architectures employed are described in quite detail, but I think it's necessary to have a link to the GitHub repository. Unless it was removed for anonymity, I strongly recommend adding it.

**Incremental contributions**

Could you maybe clarify how does your work relate with the one of [1]?

[1] Ma, Jing, et al. "Clear: Generative counterfactual explanations on graphs." Advances in neural information processing systems 35 (2022): 25895-25907.

**Final Rebuttal Confidence:**

4

**Final Rebuttal Justification:**

My rating was positive even before the rebuttal phase, when some of my concerns were addressed. I see that other weaknesses, pointed out by other reviewers, were also addressed. So I confirm my initial rating.

**Justification:**

The paper is already well written and clear, most of the concerns that I raised about clarity are either minor flaws, or they could be addressed easily for the camera-ready version.
In my opinion, the elements introduced in this work are novel and incremental. They could be expanded further (see for examples my questions above), but it's definitely worth sharing with the community.

---

> ### Author Rebuttal · Authors · 2024-10-22
>
> Thanks a lot for your reply. We are very happy that your overall perception of the paper is positive. We will make sure to address the typos and syntax errors which you brought to our attention, and work on the clarity (point 1-5 of the Weakness section) of the paper according to your suggestions. The updated paper will be available before the rebuttal period ends.
>
> **Robustness:** In the current setup there are two main limitations of classifiers which can be considered 1) it will have to support the computation of gradients and 2) it has to be permutation invariant. As we mainly are interested in obtaining explanations for deep neural networks the first assumption is not really a challenge. The second assumption does however put a limitation on the classifiers which can be considered, though designing graph classifiers to be permutation invariant is common practice.
>
> **Binary or Multiclass classification:** The method could also be employed for multiclass counterfactual explanations. In this case, the latent code would be updated in the direction of the desired class exactly as in the binary case. However, the datasets we consider in this paper (AIDS, Mutagenicity, and NCI1) are all very commonly used for examining methods for the generation of counterfactual explanations, and they are unfortunately, as most graph datasets made for this purpose, binary.
>
> **Evaluations based on ground truth:** We will attempt to answer this to the best of our ability, but we have some uncertainty to what you are asking about, so please let us know if further clarification is needed. The Mutagenicity dataset comes with labels on the individual molecules (mutagenic/not mutagenic), in the same way as the other datasets which we consider, and we are not entirely certain about which ground truth you are referring to, and which evaluation may be required. We would be very happy to hear any suggestions you may have regarding this. However, it is very important to note that we seek to provide counterfactual explanations for the \textit{classifier}. That is, even if there would be some ground truth molecular property which would fully determine whether a molecule is mutagenic or not, the generated counterfactuals explanations may not match this groundtruth, since these are generated in the context of the classifier.
>
> **Nearest Neighbour having low validity:** I am very happy that you ask this question, as it is a subtlety which is not obvious. As described, this method provides the graph of the nearest latent code with the desired label as a counterfactual. However, the graphs which are misclassified by the classifier often have latent codes close to the ones of the graphs of the opposite class. Therefore, when using these as counterfactuals, the validity naturally becomes lower. This is easier explained using a graphical illustration, and if there is enough space in the camera ready paper, then we can include one here.
>
> **Github repository:** We plan to include a Github repository upon acceptance. At the moment the project repository contains other unpublished work. As such we will make a separate repository containing the code for this project.
>
> **Relation to CLEAR paper:** The CLEAR paper is an alternative VAE based method where the VAE is trained with a “Counterfactual Prediction Loss”. As such counterfactuals are “predicted” directly from a factual graph. This is in opposition to our method, where we iteratively update the latent code of the factual graph, and thereby arrive at a counterfactual. This obviously has the advantage, that we do not have to train the VAE with the generation of counterfactuals in mind.

---

### Official Review · Reviewer_BpLm · 2024-10-09
**Review of paper 48**

**Confidence:** 3

**Summary:**

The authors propose a method to generate semantically meaningful counterfactual explanations for graph classifiers by traversing the well regularised latent space learned by PEGVAE. The authors validate their approach and show the its effectiveness on three graph datasets.

**Strengths:**

* The paper is extremely well written and was very interesting to read.
* The idea to generate the graph counterfactual explanations by optimally traversing the regularized latent space learned by VAE is novel.
* The results on three datasets are promising towards the applicability of the approach.

**Weaknesses:**

* How scalable is the approach to more complex real-world problems with large and sparse graphs?
* The quality of the counterfactuals is directly tied to how well the VAE's latent space reflects the relationships in the original graph, which might not always be optimal.

**Justification:**

The idea is novel, interesting and well executed with the experimentation, as well as the paper is well written.

---

> ### Author Rebuttal · Authors · 2024-10-22
>
> Thank you for your reply. We are happy that you find the ideas interesting and novel, and especially, that you find the paper easy to read!
>
> **Scalability:** As noted in the paper, we use a dense graph representation which puts a limit on the number of nodes which we can realistically handle when training the generative model, and it would be very interesting to explore alternative architectures to alleviate this problem in the future. To the best of our knowledge an architecture with properties similar to the PEGVAE but supportive of sparse graphs does not yet exist. However, our proposed method per se, of using an invariant classifier to guide the traversal of an equivariant latent space (also for other group actions, beyond permutation group action on graphs) does not have inherent scalability issues.
>
> **Quality:** Indeed the quality of the counterfactuals are directly tied to the idea that 1) the latent representation is meaningful, and 2) the generative model (decoder) is of sufficient quality. That the latent representation should be meaningful, is one reason why a variational autoencoder is chosen over for instance a diffusion model. That the quality of the generative model may not be sufficient for creating valid counterfactual graphs is not limited to our method, but also a limitation of other VAE base methods, e.g. CLEAR [1].
>
> [1] Ma, J., R. Guo, S. Mishra, and A. Zhang. 2022. “Clear: Generative Counterfactual Explanations on Graphs.” Advances in Neural Information Processing Systems.

---

### Meta-Review · Area_Chair_XwGk · 2024-11-02

**Recommendation:** Accept (Poster)
**Confidence:** 4

**Metareview:**

The authors propose a method to generate counterfactual examples of GNN decisions by pushing the output to cross the decision boundary line. Nevertheless, the reviewers and the authors qualify the results as preliminary as the method could be improved.

I, therefore, recommend accepting the paper for a poster session.

**Suggested Changes To The Recommendation:**

3: I agree that the recommendation could be moved up

---

### Decision · Program_Chairs · 2024-11-06

**Decision:**

Accept (Oral)

**Comment:**

Given the AC positive recommendation, we recommend an oral and a poster presentation given the AC and reviewers recommendations.